# Estimated total cardiovascular risk in a rural area of Bangladesh: a household level cross-sectional survey done by local community health workers

Mohammad Mostafa Zaman [1], Mohammad Moniruzzaman [2], Kamrun Nahar Chowdhury,[3] Salma Zareen,[4] AHM Enayet Hossain[5]

¹Research and Publication, WHO Country Office for Bangladesh, Dhaka, Bangladesh
²Public Health, Shiga University of Medical Science, Otsu, Japan
³Epidemiology, National Centre for Control of Rheumatic Fever and Heart Diseases, Dhaka, Bangladesh
⁴Director, National Centre for Control of Rheumatic Fever and Heart Diseases, Dhaka, Bangladesh
⁵Non-Communicable Diseases Control Programme, Directorate General of Health Services, Dhaka, Bangladesh

**Correspondence to**
Dr Mohammad Mostafa Zaman; zamanm@who.int

## ABSTRACT

**Objective** The aim of this study was to estimate 10-year cardiovascular disease (CVD) risk among Bangladeshi rural community residents, using the 2014 WHO/International Society of Hypertension (WHO/ISH) risk prediction charts.

**Study design** Cross-sectional population-based study done by local community healthcare workers engaging the lowest level facilities of the primary healthcare system.

**Setting and participants** A total of 1545 rural adults aged ≥40 years of Debhata upazila of Satkhira district of Bangladesh participated in this survey done in 2015. The community health workers collected data on age, smoking, blood pressure, blood glucose and treatment history of diabetes and hypertension.

**Primary outcome measures** We estimated total 10-year CVD risk using the WHO/ISH South East Asia Region-D charts without cholesterol and categorised the risk into low (<10%), moderate (10%–19.9%), high (20%–29.9%) and very high (≥30%).

**Results** The participants' mean age (±SD) was 53.9±11.6 years. Overall, the 10-year CVD risks (%, 95% CI) were as follows: low risk (81.6%, 95% CI 78.4% to 84.6%), moderate risk (9.9%, 95% CI 7.4% to 12.1%), high risk (5.8%, 95% CI 4.4% to 7.2%) and very high risk (2.8%, 95% CI 1.5% to 4.1%). In women, moderate to very high risks were higher (moderate 12.1%, high 6.1% and very high 3.7%) compared with men (moderate 7.5%, high 5.5% and very high 1.9%) but none of these were statistically significant. The age-standardised prevalence of very high risk increased from 2.9% (0.7%–5.2%) to 8.5% (5%–12%) when those with anti-hypertensive medication having controlled blood pressure (<140/90 mm Hg) added.

**Conclusion** The very high-risk estimates could be used for planning resource for CVD prevention programme at upazila level. There is a need for a national level study, covering diversities of rural areas, to contribute to national planning of CVD prevention.

## BACKGROUND

Cardiovascular disease (CVD) is the leading cause of deaths globally, taking an estimated 17.9 million lives in 2016, representing 31% of all deaths.[1 2] Of these deaths, 85% are due to heart attack and stroke. Over three-quarters

### Strengths and limitations of this study

► The present study is the first-ever population-based study in Bangladesh done by community health workers to identify those who need drug therapy for cardiovascular disease (CVD) prevention (using the 2014 WHO/International Society of Hypertension CVD risk prediction charts) under guidance of the local health authority.

► The use of local community health workers to conduct the survey at household level provided an inherent strength of community participation. Current study has a high potential of replication all over Bangladesh because upazila health system is similar everywhere.

► The estimates that we present here are based on South-East Asian regional criteria of the CVD risk charts in the absence of any national criteria is an inherent weakness of the approach.

► There is threat of generalisability because the survey was done in a single upazila of a country having hundreds of upazilas.

► Absence of cholesterol data and use of casual blood glucose data in the absence of classic symptoms of diabetes are limitations too.

of CVD deaths take place in low-income and middle-income countries (LMICs).[3] The WHO's South-East Asia Region has 11 countries, all of which fall under the LMIC category.

Deaths from CVDs are, to a great extent, preventable through targeting their risk factors. These risk factors include smoking, unhealthy diet, alcohol in excess, sedentary behaviour, high blood glucose and high blood pressure. Identification and management of individual risk factors need a huge effort from the health system. To help reduce the global burden of CVDs, WHO member states have committed to provide counselling and drug treatments for at least 50% of eligible people (defined as aged 40 years or older and

at high risk of CVDs) by 2025.[4] Accordingly, a total risk approach developed by WHO/International Society of Hypertension (WHO/ISH) makes the approach easier and cost-effective.[5][6]

In Bangladesh, a South-East Asian country, 30% of the total deaths were due to CVD in 2016.[2] National health system's response to identify population at high risk for CVDs is still absent.[2] The availability of trained health workforce, especially community health workers, is a prerequisite to implementing such programmes. Primary healthcare system in rural areas of Bangladesh includes fixed facility services (upazila health complex, union subcentres and community clinics), and domiciliary services provided by community health workers. The community clinics (total 13 000) are the lowest level and most peripheral facilities catering services to approximately 6000 people of surrounding area.[7] These clinics are run by community healthcare providers for catering services for selected communicable diseases.[7] They screen patients for hypertension and diabetes and treat them under the guidance of doctors working in the union subcentres and upazila health complexes. However, their linkage to the community health workers are not well established.

Our previous study showed that the basic package for essential NCD (PEN) intervention for non-communicable disease (NCD) prevention, especially CVD, is a feasible and realistic option at primary healthcare level.[8] However, the proportion of rural people who are at high risk of CVDs are not known. The community health workers' capacity to contribute to CVD prevention at community level is also unknown. Their engagement in a low resource setting like ours might substantially contribute to primary prevention efforts for screening, counselling and community awareness. Therefore, we aimed at conducting a household level assessment by community health workers to determine the proportion of population at high-risk for CVD in the catchment area of community clinics of a selected upazila in Bangladesh.

## METHODS

### Study design and setting

Bangladesh has 492 upazilas (as of 19 December 2017), which is the second lowest tier of regional administration in Bangladesh after a district (total 64). One upazila on average has eight unions. Debhata is one of the upazila of Satkhira district located in south-western part of Bangladesh. Its projected population for 2015 was 132 303. The adults of Debhata have 5 years of median schooling,[8] which is classical of Bangladeshi rural area. We selected Debhata upazila because the country's first NCD intervention was implemented there in 2013–2014.[8]

Experienced community health workers collected data at household level from August to December 2015 in Debhata upazila. Participants were resident adults aged 40 years or older residing in the catchment areas of all 16 community clinics of the upazila. These 16 community

clinics were located in 16 of 57 'mauza' (the primary sampling units of rural area designated by the statistical authority of Bangladesh). All these community clinics had a list of households in their respective 'mauza'. In each mauza, community health workers selected 100 eligible residents consecutively starting from the first household in the list and continued visiting households until 100 could be selected. Thus, the total sample size was 1600. At least two home visits were done to keep the non-response minimum. As usual, several people were not present at home during data collectors' visit. Refusal was almost absent. Before deployment for the data collection, 16 community health workers from their respective area were trained for 3 days. They were given a set of equipment: analogue weighing scale, flexible metallic tape, aneroid sphygmomanometer, stethoscope and glucometer. A supervised dry-run of the data collection was done on the fourth day.

### Patient and public engagement

There was no patient involvement. However, participants who needed treatment were referred to nearby community clinic for further evaluation and referral if needed. Members of the public were engaged through the management committee of the community clinics and local schools, and the union councils before launching the initiative.

### Data collection

The community health workers interviewed subjects, measured height, weight and causal capillary blood glucose. The interview included information on age, sex, education, occupation, tobacco use, treatment for hypertension and diabetes and history of CVD event. The questionnaire was in Bangla, adapted from the national STEPS survey 2010.[9] Height and weight were measured for calculating body mass index (kg/m$^2$). Blood pressure was measured twice using an appropriately sized arm cuff while the participants were in seating position. An aneroid sphygmomanometer (ALRK, Japan) was used. The first measurement was performed after 5 min of rest, and the second measurement was taken 2 min after the first measurement. We used the second measurement values in the analyses. Capillary blood glucose irrespective of prandial status[10] was measured using a glucometer (Accu-Chek Softclix CE0088, Roche Diabetes Care, Germany) on the right index finger avoiding much stasis.

### Estimation of 10-year CVD risk

We estimated 10-year CVD risk using 2014 WHO/ISH risk prediction charts (hereinafter referred to as CVD risk chart) for South East Asia Region-D without cholesterol[5] for a fatal or non-fatal major cardiovascular event. The variables used for this estimation were age (40–49 years, 50–59 years, 60–69 years and ≥70 years), sex (men and women), systolic blood pressure (<140, 140–159, 160–179 and ≥180 mm Hg), smoking status (smoker or non-smoker) and the presence or absence of diabetes (casual

capillary blood glucose >11.1 mmol/L or use of medication for diabetes). The 10-year CVD risk categories were low (<10%), moderate (10%–19.9%), high (20%–29.9%) and very high (≥30%).[5] Prior to analysis, weighting of the data were done to account for selection probabilities and non-response to remove bias from a survey sample and make the results better project the target population (all citizens of Debhata aged 40 years or older).[11]

### Data analysis

A total of 1600 participants (734 were men and 866 women) could be recruited by approaching 1721 consecutive eligible subjects (response rate, 93%). Among them, 55 had a history of CVD events (heart attack or stroke) and were excluded from our analysis as recommended.[5] Finally, our analysis is based on 1545 persons.

Mean (SD) for continuous variables and proportion for categorical variables were obtained. Results on risk categories were presented for sexes combined and separately. Age stratification of the risk categories were not done because it has already been taken in to account for calculating the risk. We categorised the CVD risk into low (<10%), moderate (10%–19.9%), high (20%–29.9%) and very high (≥30%).[5] Complex sample analyses were done to obtain weighted prevalence estimates and 95% CIs. The 95% CIs of the risk estimates were used to examine differences between groups, overlapping intervals were considered non-significant different at 5% level.

Furthermore, we have estimated the prevalence after adding those with antihypertensive medication having normal blood pressure (<140/90 mm Hg) to the chart-calculated proportions as per CVD risk chart practice notes for clinicians.[5 12] This approach has ensured that duplicate counts of individuals are not done for those who were classified using systolic blood pressure categories.

### RESULTS
### Background characteristics

Results are based on 697 men (45.1%) and 848 women (54.9%) (table 1). Their mean age was 53.9 years, with a SD of 11.6 years. The number of people persistently decreased from (42.4%) in the youngest age group (40–49 years) to 13.9% in the oldest group (70 years or older). Less than one-quarter (24.3%) of them completed primary education. The majority (86.1%) of women were housewives, and the most common occupation in men was agriculture and other manual works (45.8%).

Smoking was almost absent in women (1.2%) but common in men (36.6%). Most participants had systolic blood pressure <140 mm Hg (76%), which gradually declined thru intervening categories to 1.6% in the ≥180 mm Hg category. More than one-third (36%) people had hypertension (blood pressure ≥140/90 mm Hg or history of medication), and 1 in 10 (9.4%) had diabetes (capillary blood glucose >11.1 mmol/L or medication).

### Distribution of 10-year CVD risk

All subsequent analyses were weighted. The distribution of 10-year CVD risk in the study participants are presented for sexes combined and separately (table 2). Most participants had a low risk of CVD (81.5%, 95% CI 78.4% to 84.6%), while the moderate, high and very high risk were 9.9% (95% CI 7.4% to 12.1%), 5.8% (95% CI 4.4% to 7.2%) and 2.8% (95% CI 1.5% to 4.1%), respectively. In total 18.5% had moderate to very high level of risk. In general, women had relatively high prevalence of moderate, high and very high risk compared with men, although statistically non-significant as indicated by overlapping 95% CIs.

A subgroup analysis indicated that 10.9% people of low risk group had high systolic blood pressure (≥140 mm Hg). This persistently increased to 97.5% in the very high risk group (figure 1). However, these proportions were relatively small in case of high blood glucose (≥11.1 mmol/L), 4.8%–31.7% (figure 1). Obesity (body mass index ≥30 kg/m$^2$) also showed an increasing trend across CVD risk categories (results not shown).

Age-standardised results of very high CVD risk across sex groups are presented in table 3. The prevalence was 2.9%. Men had lower prevalence (1.9%) compared with women (4.4%). We added proportion of people with self-reported medication history for hypertension having controlled blood pressure (<140/90 mm Hg), the overall proportion increased to 8.5% (6.5% in men and 11.1% in women).

### DISCUSSION

The present study is the first-ever population-based study in Bangladesh to estimate the 10-year CVD risk scores done by community health workers. In this household survey of people aged 40 years or older living in a rural area of Bangladesh, we found that 2.9% had very high CVD risk who require medication for treatment. Addition of history of medication for hypertension (and controlled blood pressure, <140/90 mm Hg) increased this to 8.5%. These findings underscore the importance of pharmacological interventions in a cost-effective manner and use of community health workers for CVD prevention and control measures.

The risk charts were developed for the LMICs,[6] which is pertinent for Bangladeshi population. In a representative sample of an upazila, we found a low risk for the 10-year CVD risk in majority (81.5%) of the adults aged ≥40 years living in a rural area. This estimate is comparable to South-East Asia regional estimate,[13] and neighbouring Asian countries such as Pakistan (79.2%),[14] Nepal (86.4%),[15] Sri Lanka (86.4%)[16] and India (82.7%).[17] These, however, should be interpreted with caution because of differences in age of the subject, which is an important determinant. Another point is the version of the chart. A few previous studies used 2007 version, we used 2014 version. Therefore, our results may not be directly comparable without considering the chart version and age group of

**Table 1** Characteristics of the study participants, Bangladeshi rural adults aged 40 years or older years, results are number (per cent)

| Characteristics | Total (n=1545) | Men (n=697) | Women (n=848) |
|---|---|---|---|
| Age in years, mean (SD) | 53.9 (11.6) | 56.4 (11.7) | 51.8 11.2) |
| Age categories | | | |
| 40–49 | 655 (42.4) | 229 (32.9) | 426 (50.2) |
| 50–59 | 425 (27.5) | 209 (30.0) | 216 (25.4) |
| 60–69 | 250 (16.2) | 130 (18.7) | 120 (14.2) |
| ≥70 | 215 (13.9) | 129 (18.5) | 86 (10.1) |
| Education, above primary | 375 (24.3) | 247 (35.4) | 128 (15.1) |
| Occupation | | | |
| Business (small or big) | 232 (15.0) | 228 (32.4) | 4 (0.5) |
| Agriculture and other manual works | 358 (23.2) | 322 (45.8) | 36 (4.3) |
| Household works | 749 (48.4) | 21 (3.0) | 728 (86.1) |
| Others* | 207 (13.4) | 132 (18.8) | 75 (8.9) |
| Smoking tobacco | 265 (17.2) | 255 (36.6) | 10 (1.2) |
| Obesity, body mass index ≥30 kg/m$^2$ | 155 (7.4) | 40 (5.7) | 75 (8.8) |
| Systolic blood pressure categories, mm Hg | | | |
| <140 | 1174 (76.0) | 571 (81.9) | 603 (71.1) |
| 140–159 | 257 (16.6) | 95 (13.6) | 162 (19.1) |
| 160–179 | 85 (5.5) | 23 (3.3) | 61 (7.3) |
| ≥180 | 27 (1.8) | 6 (0.9) | 21 (2.5) |
| Diastolic blood pressure categories, mm Hg | | | |
| <90 | 1109 (71.8) | 528 (75.6) | 581 (68.5) |
| 90–99 | 263 (17.2) | 113 (16.2) | 150 (17.7) |
| ≥100 | 173 (11.2) | 56 (8.0) | 117 (13.8) |
| Medication for | | | |
| Hypertension | 225 (14.6) | 72 (10.3) | 153 (18.0) |
| Diabetes | 111 (7.2) | 51 (7.3) | 60 (7.1) |
| Hypertension† | 556 (36.0) | 206 (29.6) | 350 (41.3) |
| Diabetes‡ | 141 (9.1) | 65 (9.3) | 76 (9.0) |

*Includes: salaried work, unemployed, weaver, beggar, cook, tailor, cobbler and others.
†Blood pressure ≥140/90 mm Hg, or use of antihypertension medication.
‡Random capillary glucose ≥11.1 mmol/L, or use of antidiabetes medication.

the subjects. Three studies in Bangladesh reported CVD risk in adults.[18–20] Two of them were done in rural populations. The proportion of people who had high risk (≥20%) varied largely from 2.1%[19] to 10.0%[20] between two studies. Therefore, a conclusive prevalence estimate for rural area is yet to be drawn for Bangladesh. There might be community-specific features to influence the risk estimates.

**Table 2** Per cent (95% CI) people having categories of 10-year CVD risk (fatal and non-fatal)*

| CVD risk categories | Men and women (n=1545) | Men (n=697) | Women (n=848) |
|---|---|---|---|
| Low (<10%) | 81.5 (78.4 to 84.6) | 85.0 (80.4 to 89.6) | 78.2 (74.4 to 81.9) |
| Moderate (10%–19.9%) | 9.9 (7.4 to 12.1) | 7.5 (5.2 to 9.9) | 12.1 (9.0 to 15.2) |
| High (20%–29.9%) | 5.8 (4.4 to 7.2) | 5.5 (3.4 to 7.6) | 6.1 (4.5 to 7.6) |
| Very high (≥30%) | 2.8 (1.5 to 4.1) | 1.9 (0.4 to 2.5) | 3.7 (1.9 to 5.5) |

*Using the WHO/ISH South East Asia Region-D charts without cholesterol.
CVD, cardiovascular disease ; ISH, International Society of Hypertension.

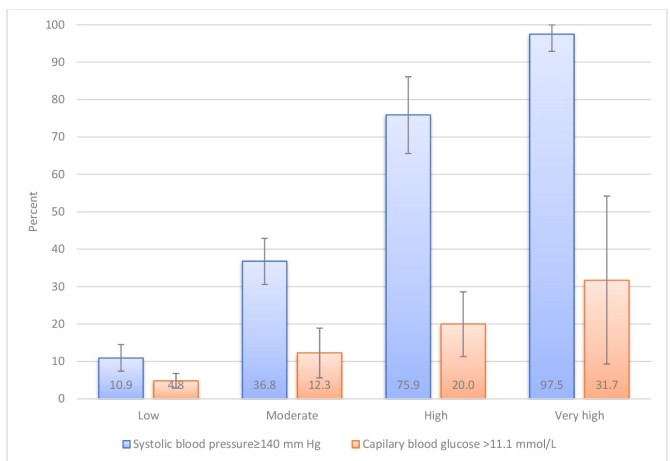

**Figure 1** Weighted proportion of people with high systolic blood pressure (≥140 mm Hg) and high blood glucose (≥11.1 mmol/L) across four 10-year cardiovascular disease risk categories* according to WHO/ISH risk scores (n=1545). *Low risk indicates <10%; moderate risk, 10%–19.9%; high risk, 20%–29.9%; very high risk, ≥30%; error bars indicate 95% CIs. ISH, International Society of Hypertension.

Out-of-pocket healthcare expenditure in Bangladesh is very high (60%), and most of this expenditure is for medicines.[21] We believe that the use of 10-year CVD risk would decrease the out-of-pocket and other costs[14] although this has a potential of undertreatment. The total risk approach may help reducing the patients' load to the health system too, which is already overburdened in Bangladesh. Because the community health workers can detect CVD risk at household level, the coverage of primary healthcare for the prevention of CVD events would be more efficient if they are used.

Despite a large proportion of deaths due to CVDs in Bangladesh,[22] preventive services for CVDs in Bangladesh are limited. Only 16% of healthcare facilities (ie, hospitals, community clinics) had the resources to diagnose, treat and manage patients with CVDs.[23] The upazila health complexes provide both preventive and curative services. Next to the upazila health complexes the union level health and family welfare centres cater services to people through doctors, medical assistants and family welfare visitors depending on their availability. The ultimate level of healthcare facilities in rural areas of Bangladesh are the community clinics at the door steps of people. The

services at these facilities are limited to the measurement of blood pressure or referrals only.[23] These centres can provide better services if these can be integrated into the domiciliary services provided by the community health workers. The current study provides evidences, for the first time, that health workers can screen out potential CVD risk for further evaluation at next level. Considering relative complexities of the CVD risk estimation methods, hypertension detection and treatment at community level could be an entry point because 97.5% of those who must get treatment (very high risk group) have hypertension. Three-quarter (75.9%) of the high risk group also can be brought under treatment using blood pressure screening at community.

Although we provide evidence that community healthcare workers can contribute to detection of people with a high risk of CVD for initiating treatment at primary healthcare facilities by physicians, they need to be trained well. Further studies are warranted to study more on the cut-off points (20% or 30%) of CVD risk charts for initiating medicines. Both these cut-off points have potentials of overdiagnosis and underdiagnosis. Nonetheless, most studies favour a 30% cut-off point for CVD risk estimation. WHO has recommended that at least 80% of health facilities should have the availability of affordable basic technologies and essential medicines necessary to treat significant NCDs, including CVDs.[4] Screening of rural adults by community health workers could contribute substantially to achieve this target.

The number of people who needs these services based on single risk factor would be huge to deal by a limited number of health service providers where supplies and medicines remain scarce. By integrating the CVD risk prediction charts into the national guidelines for the management of NCDs, the resource needs could be easily reduced without compromising the quality of services to those who are at high risk of CVDs. The implementation of CVD risk prediction chart may become popular if adequate supply of essential medicines is ensured, which is currently inadequate,[8] to fixed facilities where they will be referred. The use of local health assistants to conduct the survey at household level, a weighted analysis for the local upazila and age-standardisation for Bangladeshi population at large provided an inherent strength of community participation. Considering the similarity of the environment around the community clinics the

**Table 3** Age-standardised* prevalence (95% CI) of people aged ≥40 years with very high CVD risk ≥30% using the WHO/ISH chart plus anti-hypertensive medication†

| CVD risk categories | Men and women (n=1545) | Men (n=697) | Women (n=848) |
|---|---|---|---|
| Simple use of chart | 2.9 (0.7 to 5.2) | 1.9 (0.2 to 4.1) | 4.4 (1.0 to 8.5) |
| Chart plus medication for hypertension† | 8.5 (5.0 to 12.0) | 6.5 (1.9 to 11.1) | 11.1 (5.8 to 16.5) |

*Standardised for Bangladesh population age structure of census 2011.
†For only those used medication and had controlled blood pressure (<140/90 mm Hg). Others were already accounted for by systolic blood pressure criteria.
CVD, cardiovascular disease ; ISH, International Society of Hypertension.

current study has a high potential of replication all over Bangladesh.

Our study had limitations too. The estimates that we present here are based on South-East Asian regional criteria of the CVD risk charts in the absence of any national criteria. There might be between-country difference in applicability of the prediction charts. Besides, the study was done in a single upazila, not in Bangladesh as a whole. Absence of cholesterol data and use of casual blood glucose data in the absence of classic symptoms of diabetes are limitations too. Further, our data were collected back in 2015, and we cannot analyse out data using 2019 CVD risk chart[13] because the control measures were already planned based on our findings. We presume that the data of CVD risk using the 2019 charts will be different than what we report here.[24]

## CONCLUSION

About one in twelve rural Bangladeshi people aged 40 years or older need treatment for CVD prevention. Despite having a potential of undertreatment, this information can be used for planning resource for CVD prevention considering resource constraints. The use of community health workers to estimate their 10-year CVD risk could be an option in Bangladesh. This approach has a potential for better coverage of essential healthcare for those who are at a very high risk of CVD events. There is a need for national level study covering diversities of rural areas. Future studies should consider including the obesity and total cholesterol in line with 2019 CVD prediction chart.

**Acknowledgements** The authors are grateful to the field team especially the health inspectors of Debhata Upazila Health Complex for organising the training of the health assistants and collecting the data. The authors thank Reazwanul Haque Khan and Suraiya Akter of WHO for providing logistical and secretarial support to the PEN project. The health assistants of Debhata are Md Waheduj Zaman, Al-Hamra Parvin, Debashis Sardar, Munjila Khatun, Md Ashadul Hoque, Helali Begum, Supria Ghosh, Farhana Sultana, Md Tazibur Rahman, Dipa Rani Adhikary, SK Sharifuzzaman, Md Abdul Alim, Sree Sipra Rani, SK Rashekul Islam, S M Tazwddin Ahmed and Masuma Yeasmin.

**Collaborators** Omar Ali (Programme Manager) and Abdul Alim (Deputy Programme Manager) of NCD Control at Directorate General of Health Services, Dhaka; SMA Rahman (UHFPO) of Debhata Upazila Health Complex, Satkhira; and Alamgir Sikdar (Project Officer), Subrato Gosh (Field Coordinator), and Masud Ahmed (Data Manager) of the WHO-Government joint PEN Project.

**Contributors** MMZ: conceptualised the project, developed the protocol, trained manpower, analysed data, interpreted results critically and drafted the manuscript. MM, KNC, SZ: prepared the manual and trained the field team, implemented the survey in the field and revised the manuscript. AHMEH guided the work, coordinated the fieldwork and reviewed the manuscript critically. He is the guarantor of data. All authors have approved the submission.

**Funding** Financial assistance for this study was provided by the WHO Bangladesh (WHO Reference: 2015/545398–0, Purchase Order: 201288322).

**Competing interests** None declared.

**Patient and public involvement** Patients were not involved in the design, or conduct, or reporting, or dissemination plans of this research. However members of the public were encouraged to particiapate in the conduct of the field work, attend the counselling sessions through yard meetings, and receive treatment when needed. Refer to the Methods section for further details.

**Patient consent for publication** Not required.

**Ethics approval** This survey was done by the Upazila Health System to see whether the community workers can get cardiovascular disease risk factor data and provide feedbacks to the community. In addition, acceptance of household level measurement of blood pressure, etc and counselling services provided by the community health workers were examined. Community consents were obtained engaging the management committee of the community clinics, union councils and management committees of all schools before launching the initiative. These were in line with the ethics committee approval of the National Centre for Control of Rheumatic Fever and Heart Diseases (No. NCCRFHD/2015/01). The Committee approved getting the consent verbally because this assessment was a health service-oriented work as per the mandate of the Upazila Health Complex. Participants had the rights to withdraw at any stage of the assessment and not to participate at all. Participants were given a report of their risk factor status and referral was done for treatment if necessary. All the health workers of the upazila irrespective of their involvement in the assessment were trained so that the health benefits reach out all the households of Debhata upazila.

**Provenance and peer review** Not commissioned; externally peer reviewed.

**Data availability statement** Data are available upon reasonable request. Please contact Dr MM Zaman at: zamanm@who.int.

**ORCID iDs**
Mohammad Mostafa Zaman http://orcid.org/0000-0002-1736-1342
Mohammad Moniruzzaman http://orcid.org/0000-0003-2144-7111

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
