## [Reviewer comments · BMJ Open]

ARTICLE DETAILS

TITLE (PROVISIONAL)	Estimated total cardiovascular risk in a rural area of Bangladesh: A household level cross-sectional survey done by local community health workers
AUTHORS	Zaman, MM; Moniruzzaman, Mohammad; Chowdhury, Kamrun; Zareen, Salma; Hossain, A.H.M. Enayet

VERSION 1 – REVIEW

REVIEWER	Ene-Iordache, Bogdan Istituto di Ricerche Farmacologiche Mario Negri IRCCS, Biomedical Engineering
REVIEW RETURNED	01-Dec-2020

GENERAL COMMENTS	General Comments In this manuscript, Zaman et al. present a cross-sectional study aimed at evaluating the 10-year cardiovascular (CVD) risk in a rural population of Bangladesh. To evaluate the 10-year CVD risk they employed the 2014 WHO Charts for calculating CVD risk without cholesterol, applied to 1,545 subjects aged 40 or older. Data collection and clinical parameter measurements were performed by community health operators at the household level. By applying merely WHO charts, the authors estimated a relatively low prevalence of CVD high risk (2.8%), but this estimate increased up to 13.1% if blood pressure $\geq 160/100$ mmHg, and then up to 21.2% if treatment for hypertension was considered. The authors conclude that such estimates could be used for resource planning for CVD prevention programs at regional level, and their study might be replicated at national level in Bangladesh. The authors have attempted to carry out a study in a setting where resources may be quite limited, and for this, they should be congratulated. The manuscript, however, must be potentially improved in a major revision before getting acceptance in BMJ Open. Major Comments One major point in my opinion is the lack of listing all study collaborators. Since the title clearly states that the study was “done by local community health workers”, I strongly suggest to include the
---

list of all health workers as well as all other staff involved in the study (e.g. investigators, research nurses, data monitoring, data managers, laboratory staff, etc.). Such groups are usually included in publications as “<Study ACRONYM> Organization” and their members listed also in Pubmed as “collaborators”. Please find below a BMJ Open reference that you can browse in Pubmed, it is a clear example of what I intend with this list of collaborators:

Lakin JR, Brannen EN, Tulsy JA, Paasche-Orlow MK, Lindvall C, Chang Y, Gundersen DA, El-Jawahri A, Volandes A; ACP-PEACE Investigators. Advance Care Planning: Promoting Effective and Aligned Communication in the Elderly (ACP-PEACE): the study protocol for a pragmatic stepped-wedge trial of older patients with cancer. *BMJ Open*. 2020 Jul 14;10(7):e040999. doi: 10.1136/bmjopen-2020-040999.

Page 6, line 161: in my opinion reference 9 cannot be cited here.

Table 2: for clarity, in the first column add the categories of 10-years CVD risk, i.e. low, moderate, high, very high.

In my opinion, obesity categories should be better defined. Bangladesh is an Asian country for which the WHO consultation (WHO Expert Consultation. Appropriate body-mass index for Asian populations and its implications for policy and intervention strategies. *Lancet* 2004; 363: 157–63.) recommended for obesity BMI 27.5 kg/m² or higher.

Table 3 and related text: since recalculating the very high risk (>30%) category it will change appropriately all other three categories, it would be better to recalculate the new risk including BP & hypertension medication and present all categories as in Table 2. Once obtained these new variables, logistic regression would be a better test than merely Figure 1 to investigate the possible association between the risk factors (low vegetable & fruit intake, sedentary behavior, obesity) and the 10-year CVD risk.

Figure 1 does no more than illustrating that subjects with lower prevalence of CVD risks are more likely to have lower risks (not related to CVD risk calculation), which is well-known.

Minor Comments

	Abstract, line 43: "blood pressure" is listed twice as data collection. Abstract, lines 48-49: "the a 10-year CVD risk (% , 95% CI) were as follows:" should be "the 10-year CVD risks (% , 95% CI) were as follows:" (if plural) or "the 10-year CVD risk (% , 95% CI) was as follows:" (if singular). Abstract lines 49-50: prevalence of 10-year CVD risk should be expressed in percent (%). Please check the format to all references; for ref #5 do not provide the link to the website.
--	---

REVIEWER	Zhao, Jiaying The Australian National University, National Centre for Epidemiology & Population Health
REVIEW RETURNED	24-Dec-2020

GENERAL COMMENTS	The manuscript estimated the 10-year cardiovascular disease (CVD) risk in a Bangladeshi rural community, based on a cross-sectional population based survey. The paper is generally well written. However, the following points may be considered and revised by authors before it can be published. Main comments:  1. Sample representativeness of the local population, and potential bias between the sample and rural Bangladesh The authors should compare the demographic profile (in Table 1) with the local population. 2. Age effects: Standardisation may be considered Age is a major predictor for CVD risk by 2014 WHO/ISH risk prediction charts. However, the crude percentage of people in each CVD risk categories was highly related to age structure of the population. Standardised percentages (using the local population structure as the standard population) may be considered to report. 3. Conclusion Page 11, Line 299-Line 300 : The author concluded "that the implementation of a total risk approach for preventing CVDs engaging 300 community health workers and community clinics is a feasible option in Bangladesh." I am not so sure how the author can establish this causal relationship in the conclusion from the current results (i.e. CVD risk estimates) from a cross-sectional survey, I think that this is probably a discussion point for the paper. Minor comments:  1. Page 4, Line 103, NCD: please give the full spellings (non-communicable disease) in the first time 2. Page 5, Line113 -118: the font is different from other text.
--

REVIEWER	Kaur, Prabhdeep National Institute of Epidemiology, Division of Noncommunicable Diseases
REVIEW RETURNED	24-Dec-2020

GENERAL COMMENTS	Good data set , further analysis as suggested will make it more useful for treatment policies Overall comments
--

	 • It is a good dataset but in depth analysis should be done to make it useful for policy decisions. • One of the questions which can answered is if it is suitable to use risk stratification OR cut offs of BP/ glucose recommended by various professional societies can be used for making treatment decisions. • Who will be left out of treatment despite high BP/Glucose if risk charts are used? • Given high CVD premature mortality, are risk charts the best option? Introduction  • Explain the relevance of risk prediction charts and if there is any literature on usefulness Methods  • Organise the information under appropriate sub headings for better understanding and flow – Study design and population, sample size and sampling strategy, data collection, data analysis, ethics etc. • Design – it is not clear if the study has primary data collection or analysis of routinely collected data during service delivery • Selection of individuals – were all individuals in the selected household surveyed? • Patient and public engagement – patient involvement should be mentioned and also what was done to ensure referral and treatment of people with risk level requiring treatment • Operational definitions of risk factors not given – HT, low fruit/veg intake, sedentary etc. • Devices  ○ BP monitor : Was it a validated monitor? Company? model? ○ Height/weight/Capillary glucose – Company? Model? Results  • Table 1 – Mention % treated for HT and DM • Table 2 – Sub group analysis can be done to see what % in each risk category had BP (e.g. 140/90 mmHg) or glucose (≥ 126 mg/dL) above cut off which can be used to initiate treatment • Figure – y axis not labelled, defs of low/mod/high/very high not mentioned anywhere; component bar chart might be more suitable • Table 3 – If systolic BP is already included in the risk estimation, then why authors want to add BP as a separate variable? Are some individuals counted twice by adding BP? • If authors want to emphasise the important of treating blood pressure, they should compare what % will be treated if risk charts are used and what % will be
--	--

	treated if BP cut off of 140/90 is used. Discussion  • Cost argument in para 2 is not relevant here as no data regarding cost is presented • Discussion should focus on 2-3 messages which may emerge from additional analysis
--	--

VERSION 1 – AUTHOR RESPONSE

Reviewer: 1

Dr. Bogdan Ene-lordache, "Mario Negri" Institute for Pharmacological Research

In this manuscript, Zaman et al. present a cross-sectional study aimed at evaluating the 10-year cardiovascular (CVD) risk in a rural population of Bangladesh. To evaluate the 10-year CVD risk they employed the 2014 WHO Charts for calculating CVD risk without cholesterol, applied to 1,545 subjects aged 40 or older. Data collection and clinical parameter measurements were performed by community health operators at the household level. By applying merely WHO charts, the authors estimated a relatively low prevalence of CVD high risk (2.8%), but this estimate increased up to 13.1% if blood pressure $\geq 160/100$ mmHg, and then up to 21.2% if treatment for hypertension was considered. The authors conclude that such estimates could be used for resource planning for CVD prevention programs at regional level, and their study might be replicated at national level in Bangladesh.

The authors have attempted to carry out a study in a setting where resources may be quite limited, and for this, they should be congratulated. The manuscript, however, must be potentially improved in a major revision before getting acceptance in BMJ Open.

	Major Comments	
1	One major point in my opinion is the lack of listing all study collaborators. Since the title clearly states that the study was “done by local community health workers”, I strongly suggest to include the list of all health workers as well as all other staff involved in the study (e.g. investigators, research nurses, data monitoring, data managers, laboratory staff, etc.). Such groups are usually included in publications as “<Study ACRONYM> Organization” and their members listed also in Pubmed as “collaborators”. Please find below a BMJ Open reference that you can browse in Pubmed, it is a clear example of what I intend with this list of collaborators:	List of collaborators added Lines: 352-356
2	Page 6, line 161: in my opinion reference 9 cannot be cited here.	Reference 9 removed from here
3	Table 2: for clarity, in the first column add the categories of 10-years CVD risk, i.e. low, moderate, high, very high.	Added
4	In my opinion, obesity categories should be better defined. Bangladesh is an Asian country for which the WHO consultation (WHO Expert Consultation. Appropriate body-mass index for Asian populations and its implications for policy and intervention strategies. Lancet 2004; 363: 157–63 PubMed .) recommended for	The concerned Figure having data on BMI is removed as per your suggestion given on the Figure (point 6 below). Therefore, a change in the cut-off point of BMI is not necessary. We added obesity

	obesity BMI 27.5 kg/m ² or higher.	prevalence in Table 1 as background data only. Cut-off point remained same as has been done in most studies in Bangladeshi population.
5	Table 3 and related text: since recalculating the very high risk (>30%) category it will change appropriately all other three categories, it would be better to recalculate the new risk including BP & hypertension medication and present all categories as in Table 2. Once obtained these new variables, logistic regression would be a better test than merely Figure 1 to investigate the possible association between the risk factors (low vegetable & fruit intake, sedentary behavior, obesity) and the 10-year CVD risk.	The purpose of Table 3 is to provide simple information for the primary health care managers in a about the need for medication in a succinct manner. This is the main message of the paper too. Therefore, we humbly do not want to make it complex but present standardized estimates as per suggestion of another reviewer. Our objective is not to identify factors associated with estimated risk categories. Therefore, we humbly wish not to do logistic regression.
6	Figure 1 does no more than illustrating that subjects with lower prevalence of CVD risks are more likely to have lower risks (not related to CVD risk calculation), which is well-known.	Figure 1 is removed but the relevant text in the Results section is kept with a little modification in line with your suggestion. However, considering the potential implication of BMI in future studies using 2019 chart definitions, BMI is presented in Table 1.
Minor Comments		
7	Abstract, line 43: "blood pressure" is listed twice as data collection.	Duplicate words removed.
8	Abstract, lines 48-49: "the 10-year CVD risk (% , 95% CI) were as follows:" should be "the 10-year CVD risks (% , 95% CI) were as follows:" (if plural) or "the 10-year CVD risk (% , 95% CI) was as follows:" (if singular).	Revised to plural form.
9	Abstract lines 49-50: prevalence of 10-year CVD risk should be expressed in percent (%).	% mark inserted to all such results
10	Please check the format to all references; for ref #5 do not provide the link to the website.	Weblink removed
Reviewer 2: Dr. Jiaying Zhao, The Australian National University The manuscript estimated the 10-year cardiovascular disease (CVD) risk in a Bangladeshi rural community, based on a cross-sectional population based survey. The paper is generally well written. However, the following points may be considered and revised by authors before it can be published.		
Main comments:		
1	Sample representativeness of the local population, and potential bias between the sample and rural Bangladesh The authors should compare the demographic profile (in Table 1) with the local population.	We have weighted our sample for the population of the Debhatta Upazila aged 40 years or older to have representativeness of the local population (Lines 174-177). In addition, we added a comment on this issue in the Discussion section (Lines: 308-309).
2	Age effects: Standardization may be considered Age is a major predictor for CVD risk by 2014 WHO/ISH risk prediction charts. However, the crude percentage of people in each CVD risk categories was highly related	Results of Table 3 is standardized for age structure of Bangladeshi rural population using Bangladesh Bureau of Statistics data.

	to age structure of the population. Standardized percentages (using the local population structure as the standard population) may be considered to report.	
3	Conclusion Page 11, Line 299-Line 300 : The author concluded “that the implementation of a total risk approach for preventing CVDs engaging community health workers and community clinics is a feasible option in Bangladesh.” I am not so sure how the author can establish this causal relationship in the conclusion from the current results (i.e. CVD risk estimates) from a cross-sectional survey, I think that this is probably a discussion point for the paper.	Thank you for the suggestion. We have revised our Conclusion to “One in five rural Bangladeshi people aged 40 years or older need treatment for CVD prevention. The use of community health workers to estimate their 10-year CVD risk could be option in Bangladesh. The “very high-risk” estimates could be used for planning resource for CVD prevention programme..”
	Minor comments:	
4	Page 4, Line 103, NCD: please give the full spellings (non-communicable disease) in the first time	Full spelling added
5	Page 5, Line113 -118: the font is different from other text	Font revised
Reviewer 3: Dr. Prabhdeep Kaur, National Institute of Epidemiology		
Good data set, further analysis as suggested will make it more useful for treatment policies		
	Overall comments	
1	 It is a good dataset, but in-depth analysis should be done to make it useful for policy decisions. One of the questions which can answered is if it is suitable to use risk stratification OR cut offs of BP/ glucose recommended by various professional societies can be used for making treatment decisions. Who will be left out of treatment despite high BP/Glucose if risk charts are used? Given high CVD premature mortality, are risk charts the best option? 	Thank you for the appreciation! The data originate from the works of primary care workers guided by local level health managers. The analysis is intended for people working in primary care so that they can replicate the work. Therefore, in-depth analysis might appear cumbersome and complex to them. Conventional approach of treating patients for BP and glucose and others leads to a huge burden to the resource constrained system. Therefore, the global risk approach is used. However, debate can continue whether the use of risk-chart is the best or better option.
	Introduction	
2	Explain the relevance of risk prediction charts and if there is any literature on usefulness	We have revised the texts given in the second paragraph to highlight this. The reference to this is changed also (Lines 88-91).
	Methods	
3	Organise the information under appropriate sub headings for better understanding and flow – Study design and population, sample size and sampling strategy, data collection, data analysis, ethics etc.	We organized the texts so that it appears as a good storytelling, which has been appreciated by two reviewers above. Therefore, we humbly wish to maintain the original story. However, the points that you found unclear, are addressed as

		below.
4	Design – it is not clear if the study has primary data collection or analysis of routinely collected data during service delivery.	Primary data collected for this study. Relevant text in 130-134 revised to make it clearer.
5	Selection of individuals – were all individuals in the selected household surveyed?	Yes, all were selected in their households as mentioned in lines 130-132.
6	Patient and public engagement – patient involvement should be mentioned and also what was done to ensure referral and treatment of people with risk level requiring treatment	Those categorized as very high risk were referred to the community clinics and subsequent pathways, if needed, were followed as per the practice of the Upazila Health system. See lines 148-149.
7	Operational definitions of risk factors not given – HT, low fruit/veg intake, sedentary etc.	Hypertension has been defined upon its appearance in line 215 and footnote of Table 1. Now the Figure on low fruit/veg intake and sedentary is removed. Therefore, the definition is not needed.
8	Devices o BP monitor : Was it a validated monitor? Company? model? o Height/weight/Capillary glucose – Company? Model?	Aneroid sphygmomanometer (ALRK, Japan), weighing scale (Tanita, Japan), blood glucose (Accu-Check Softclix, Germany) have been mentioned (Lines 159-163).
	Results	
9	Table 1 – Mention % treated for HT and DM	Added
10	Table 2 – Sub group analysis can be done to see what % in each risk category had BP (e.g. 140/90 mmHg) or glucose (≥ 126 mg/dL) above cut off which can be used to initiate treatment	Suggested results are given in Figures 1 and 2. Relevant text added to the Results section, lines 225-229
11	Figure – y axis not labelled, defs of low/mod/high/very high not mentioned anywhere; component bar chart might be more suitable	These were given in the Figure legends. However, Figure has been dropped as suggestion of Reviewer 1.
12	Table 3 – If systolic BP is already included in the risk estimation, then why authors want to add BP as a separate variable? Are some individuals counted twice by adding BP?	Dropped it but kept history of medication (but only for those who had controlled BP<140/90). Ensured that people are not counted twice.
13	If authors want to emphasise the important of treating blood pressure, they should compare what % will be treated if risk charts are used and what % will be treated if BP cut off of 140/90 is used.	This has been reported in so many articles in the past, which was the basis for advocating the sue of the Chart. Therefore, we are not reporting it here.
	Discussion	
14	Cost argument in para 2 is not relevant here as no data regarding cost is presented	The cost argument is given in paragraph 3. Cost reduction by treating those who are at high risk is one of the purposes of using Charts. Therefore, we want to keep it. However, the last sentence on catastrophic CVD event is deleted.
15	Discussion should focus on 2-3 messages which may emerge from additional analysis	Dropped a few sentences to reduce the length of the paragraphs but all points are retained.

VERSION 2 – REVIEW

REVIEWER	Ene-Iordache, Bogdan Istituto di Ricerche Farmacologiche Mario Negri IRCCS, Biomedical Engineering
REVIEW RETURNED	09-Feb-2021

GENERAL COMMENTS	General Comments The Authors have satisfactorily answered to almost all my comments. The paper has been improved and I have only few minor comments on this revised version. Minor Comments Abstract, lines 53-54: delete the words “which was increased” and add the CI interval after 8.5%. New Figures/Legends Figure 1, please check the text between legend and main text: it is “...people with high blood pressure ($\geq 140/90$ mm Hg)” in the legend, and “...high systolic blood pressure (≥ 140 mm Hg)” in the main text. Also, for the sake of clarity, figures 1 and 2 can be merged in a unique one.
--

REVIEWER	Zhao, Jiaying The Australian National University, National Centre for Epidemiology & Population Health
REVIEW RETURNED	22-Feb-2021

GENERAL COMMENTS	The authors have addressed reviewers' comments well. I have only two minor comments:  1. Page 2, Line 54, may add the confidence interval “8.5% (5.0-12%)”. The grammar of the sentence may be corrected (e.g. increase from ...to ...) 2. Page 4, Line 89-91, the format of font should be changed.
---

REVIEWER	Kaur, Prabhdeep National Institute of Epidemiology, Division of Noncommunicable Diseases
REVIEW RETURNED	27-Feb-2021

GENERAL COMMENTS	Authors have addressed several suggestions however a few important suggestions are yet to be addressed.  1. What % will be treated if risk charts are used and what % will be treated if BP cut off of 140/90 is used. Given huge gap using these two approaches, it has implications for HT program planning and drug procurement policies. 2. Given high CVD premature mortality, are risk charts the best option? This need to be discussed. Proportion eligible for drugs is very low using risk charts and will leave large proportion of population untreated. South Asian population have high risk of
--

	mortality due to CVD and under treatment is a major problem. 3. Authors should give balanced recommendations and highlight the limitation of risk charts.
--	---

VERSION 2 – AUTHOR RESPONSE

Reviewer: 1

Dr. Bogdan Ene-Iordache, "Mario Negri" Institute for Pharmacological Research Comments to the Author:

General Comments

The Authors have satisfactorily answered to almost all my comments. The paper has been improved and I have only few minor comments on this revised version.

>>RESPONSE: Thank you so much!

Minor Comments

Abstract, lines 53-54: delete the words “which was increased” and add the CI interval after 8.5%.

>>RESPONSE: Corrected as per suggestion given.

New Figures/Legends

Figure 1, please check the text between legend and main text: it is “...people with high blood pressure ($\geq 140/90$ mm Hg)” in the legend, and “...high systolic blood pressure (≥ 140 mm Hg)” in the main text.

>>RESPONSE: Same texts (≥ 140 mm Hg) used in main text (line 219) and figure legend (line 449)!

Also, for the sake of clarity, figures 1 and 2 can be merged in a unique one.

>>RESPONSE: Two figures merged in to one.

Reviewer: 2

Dr. Jiaying Zhao, The Australian National University Comments to the Author:

The authors have addressed reviewers' comments well. I have only two minor comments:

>>RESPONSE: Thank you so much!

1. Page 2, Line 54, may add the confidence interval “8.5% (5.0-12%)”. The grammar of the sentence may be corrected (e.g. increase from ...to ...)

>>RESPONSE: Corrected.

2. Page 4, Line 89-91, the format of font should be changed.

>>RESPONSE: Format (font) changed.

Reviewer: 3

Dr. Prabhdeep Kaur, National Institute of Epidemiology Comments to the Author:

Authors have addressed several suggestions however a few important suggestions are yet to be addressed.

>>RESPONSE: Thank you so much!

1. What % will be treated if risk charts are used and what % will be treated if BP cut off of 140/90 is

used. Given huge gap using these two approaches, it has implications for HT program planning and drug procurement policies.

>>RESPONSE: We humbly put forward our points here. The objective of our study is to examine whether we can use our community health workers to estimate CVD risk at community level. It is not the purpose of the study to examine which approach is better. In a resource constrained society like ours the priority is screen out and provide treatment to those who need it most. It is not difficult to understand reviewer's point that treating all individual risk factors will bring more benefits especially if we target hypertension. But we take in to account that more than half of the hypertensives remain out of treatment coverage. Adding one risk factor after another will make the proportion even bigger. People cannot afford to pay the medicine cost out of their pocket, not the public health sectors will provide them full courses of medicines. Then prioritizing the issue becomes more important than scientific argument.

In response to the reviewer's comment, however, we insert a little text to highlight the potential of under-treatment of CVD risk approach and their differential coverage rates. Kindly see:

- a. Line 252 (..... although this has a potential of under-treatment).
- b. Lines 268-272: Considering relative complexities of the CVD risk estimation methods, hypertension detection and treatment at community level could be an entry point because 97.5% of those who must get treatment (very high risk group) have hypertension. Three-quarter (75.9%) of the high risk group also can be brought under treatment using blood pressure screening at community.

2. Given high CVD premature mortality, are risk charts the best option? This need to be discussed. Proportion eligible for drugs is very low using risk charts and will leave large proportion of population untreated. South Asian population have high risk of mortality due to CVD and under treatment is a major problem.

>>RESPONSE: Kindly see response to reviewer's point number 1 above. The potential of under-treatment has been emphasized in line 252. Just to repeat the arguments above, the proportion of untreated people is even larger with the current approach of treating all treatable factors/conditions such as hypertension, diabetes, (and possibly hypercholesterolemia).

3. Authors should give balanced recommendations and highlight the limitation of risk charts.

>>RESPONSE: Kindly see line number 305. The second sentence now reads: "Despite having a potential of under-treatment, this information can be used for planning resource for CVD prevention considering resource constraints."

VERSION 3 – REVIEW

REVIEWER	Kaur, Prabhdeep National Institute of Epidemiology, Division of Noncommunicable Diseases
REVIEW RETURNED	24-May-2021
GENERAL COMMENTS	Authors have addressed most of the comments. Authors may add the data in results section - What % will be treated based on BP cut off 140 Or 90 mmHg versus risk charts. This is very important for resource planning. Antihypertensives are the one of the cheapest medications and it is not appropriate to deny such a cost effective intervention just because they do not fall in high risk category in charts.